# Machine Learning-Based Definition of Symptom Clusters and Selection of Antidepressants for Depressive Syndrome

**DOI:** 10.3390/diagnostics11091631

**Published:** 2021-09-07

**Authors:** Il Bin Kim, Seon-Cheol Park

**Affiliations:** 1Department of Psychiatry, Hanyang University Guri Hospital, Guri 11923, Korea; jonecaby49@gmail.com; 2Graduate School of Medical Science and Engineering, Korea Advanced Institute of Science and Technology (KAIST), Daejeon 34141, Korea; 3Department of Psychiatry, Hanyang University College of Medicine, Seoul 04763, Korea

**Keywords:** depressive syndrome, machine learning, personalized medicine, symptom clusters, selecting antidepressants

## Abstract

The current polythetic and operational criteria for major depression inevitably contribute to the heterogeneity of depressive syndromes. The heterogeneity of depressive syndrome has been criticized using the concept of language game in Wittgensteinian philosophy. Moreover, “a symptom- or endophenotype-based approach, rather than a diagnosis-based approach, has been proposed” as the “next-generation treatment for mental disorders” by Thomas Insel. Understanding the heterogeneity renders promise for personalized medicine to treat cases of depressive syndrome, in terms of both defining symptom clusters and selecting antidepressants. Machine learning algorithms have emerged as a tool for personalized medicine by handling clinical big data that can be used as predictors for subtype classification and treatment outcome prediction. The large clinical cohort data from the Sequenced Treatment Alternatives to Relieve Depression (STAR*D), Combining Medications to Enhance Depression Outcome (CO-MED), and the German Research Network on Depression (GRND) have recently began to be acknowledged as useful sources for machine learning-based depression research with regard to cost effectiveness and generalizability. In addition, noninvasive biological tools such as functional and resting state magnetic resonance imaging techniques are widely combined with machine learning methods to detect intrinsic endophenotypes of depression. This review highlights recent studies that have used clinical cohort or brain imaging data and have addressed machine learning-based approaches to defining symptom clusters and selecting antidepressants. Potentially applicable suggestions to realize machine learning-based personalized medicine for depressive syndrome are also provided herein.

## 1. Introduction

Depression is one of the most burdensome disorders worldwide, with a lifetime prevalence of approximately 20% of the global population [1]. Depression remission after the first antidepressant trial is only 30% [2,3]. This low remission rate is partly because diagnosing depression does not guarantee heterogeneous symptom subtypes [4]. Inevitably, the concept that depression is characterized by symptomatic heterogeneity, such as atypical [5], melancholic [6], and anxious [7] subtypes, has gained considerable attention. In addition, it has been reported that the heterogeneity of depressive syndrome can theoretically result from the polythetic and operational criteria of major depression [8,9,10,11,12]. According to the *Diagnostic and Statistical Manual of Mental Disorders*, fifth edition (DSM-5) [13], a confirmed diagnosis of major depressive disorder requires both the presence of five or more symptoms among, nine symptoms, including depressed mood, diminished interest or pleasure, weight loss or gain, insomnia or hypersomnia, psychomotor retardation or agitation, fatigue or loss of energy, feelings of worthlessness or excessive guilt, diminished thinking ability or indecisiveness, recurrent thoughts of death or recurrent suicidal ideation, and the presence of either depressed mood or diminished interest or pleasure. Herein, the subset of k draws from n distinguishable objects without replacement and without regard to order that (nCk) can calculate from the theoretical number of different combinations meeting the polythetic and operational criteria of major depressive disorder in DSM-5. Thus, 227 different diagnostic symptom combinations were calculated that can fulfill the DSM-5 diagnostic criteria for major depressive disorder [14,15,16,17]. In terms of psychiatric taxonomy, the heterogeneity of depressive syndrome has been criticized by the concept of a language game in Wittgensteinian philosophy [18]. Wittgenstein suggested the analogy as follows [19]:

Consider for example the proceedings that we call games. I mean board-games, card-games, ball-games, Olympic games, and so on. What is common to them all?—Don’t say: “There must be something common, or they would not be called games”—but look and see whether there is anything common to all.—For if you look at them you will not see something that is common to all, but similarities, relationships, and a whole series of them at that. To repeat: don’t think, but look!—the concept game is a concept with blurred edges.—“But is a blurred concept a concept at all?”—Is an indistinct photograph a picture of a person at all? Is it even always an advantage to replace an indistinct picture by a sharp one? Isn’t the indistinct one often exactly what we need? (Wittgenstein, 2001).

It is also proposed that cases of depressive syndrome are conceptually related by the “family resemblance” rather than the “essence.” Thus, it is concluded that the heterogeneity of depressive syndrome is consistent with Wittgensteinian’s analogy [18]. Thus, the nomenclature of depressive syndrome can be consistent not with the categorical approach, but the dimensional approach, in the context of the heterogeneity of major depressive disorder [14]. Furthermore, based on the theoretical construct change from chemical imbalance to dysfunctional circuitry, the symptom-based approach, but not the diagnosis-based approach, has been emphasized by Thomas Insel in his work on the next generation of treatments for mental disorders [20]. Along with the heterogeneity concept, the therapeutic approach also shifts toward selecting antidepressants according to specific symptom clusters [21]. Each cluster of depression symptoms may be thought to react to specific antidepressants, thus potentially improving the current low remission rates. The theorem supporting depression heterogeneity has not generated notable clinical utility in that theory-driven classification of symptom clusters and subsequent antidepressant selection have only produced low accuracies in treatment outcome predictions [22]. However, the clinical utility of the depression heterogeneity concept in diagnostics and therapeutics is increasingly acknowledged with the use of data-driven machine learning approaches.

Machine learning approaches can be more beneficial in the study of depression compared with traditional methods. Factor analysis, for instance, may generate complicated combinations of heterogeneous symptoms within specific dimensions [23]. These analytic approaches also can be vulnerable to experimenter bias in that a researcher has to choose the number of components or clusters in data, as such in k means clustering method [24]. Hierarchical clustering, a type of machine learning method, is an easy-to-implement, deterministic approach in which each of the symptoms is assigned to a single cluster even without predetermining the desired number of clusters.

Clinical expertise in diagnostics and therapeutics currently progresses in accordance with the advent of machine learning algorithms for handling clinical big data [25]. The machine learning approach allows clinicians to consider that clinical data can be a useful source of predictors for the classification of depressive symptom clusters and selection of antidepressants. Thus, data-driven approaches, rather than clinician-based, diagnostics, and therapeutic approaches, are now incorporated into depression research to disentangle the questions as to what subtypes of depression exist, and how these subtypes react to different antidepressants. However, from a cautious perspective, clinical expertise can still be a keystone in machine learning-based diagnostics and therapeutics in depression, as clinical knowledge and experience are inevitable not only for choosing clinical data input to algorithms but also for interpreting the algorithm prediction outcomes. The next generation of machine learning-based depression practice might be dependent on the degree to which clinical expertise is fully incorporated into unmanned learning approaches to future precision medicine [26]. Thus, clinicians are required to not only be updated of the latest findings from the machine learning-based depression research but also yield clinical expertise-based perspectives for the learning algorithms in terms of better translation of the machine learning findings to clinical applications.

This review is not intended to be systematic or comprehensive for all relevant studies, but rather narrative for emphasizing predominant papers which seem of practical interest for readership. In particular, this review focuses on the studies in which machine learning-based prediction models for treatment outcome in depression were built in large clinical cohorts with several hundreds of subjects, including the Sequenced Treatment Alternatives to Relieve Depression (STAR*D) [27], Combining Medications to Enhance Depression Outcome (CO-MED) [3], and German Research Network on Depression (GRND) [28]. These cohorts encompass accessible clinical data that offer benefits such as cost-effectiveness, compared to the genetics and imaging data, and interchangeable usability of variable characteristics between cohorts for the algorithm outcome validations, in part guaranteeing generalizability. This review also provides suggestions for advancing machine learning-based depression research for better symptom classification and antidepressant selection.

## 2. Data-Driven Classification of Symptom Clusters in Depression

Depression is viewed as a heterogeneous mental construct [8]. Machine learning-based analysis underlines the benefits of addressing symptom heterogeneity by subject stratification and the application of data-driven therapeutic responses instead of summed scores of clinical scales. Although previous studies have suggested a conventional clinician experience-based (theory-driven) classification of depression subtypes, the efforts resulted in poor predictive values for therapeutic outcomes for different antidepressants. For instance, Uher et al. [21] suggested three types of depression, including melancholic, atypical, and anxious features, for which therapeutic outcomes were evaluated with escitalopram and nortriptyline. However, this clinician-driven approach to grouping depression subtypes resulted in low to modest accuracy in the prediction of therapeutic outcomes. Likewise, a similar study using clinician-driven classification of depression subtypes also showed only low prediction accuracies, thus limiting clinical utility [22]. After these trials, the science of big data was incorporated into depression research, supporting the advantages of data-driven phenotypes. Indeed, a recent review suggested evidence for the considerable prognostic value of data-driven depression subtype classification [29]. These research trends are rooted in discovering clinical signatures of predictions for response of specific symptoms to specific antidepressants, and thus require established clinical data with a large number of depressive subjects. Indeed, with the increased accessibility of large databases, multivariate models exploiting clinical data have been introduced to psychiatric research in recent years [25]. The cohort data consisted of sociodemographic (sex and age), diagnostic (scale scores), and therapeutic variables (antidepressant classes), which can be assumed to be useful candidate predictors for the classification of depression subtypes. In particular, the STAR*D database in the US and the GRND and Group for Studies of Resistant Depression (GSRD) databases in Europe have facilitated progress in predicting therapeutic outcomes at each patient level [30,31,32]. Even though there are approaches of integrating more comprehensive data, including pharmacogenomics [33], in elucidating the machine learning-based study of depression, this may be beyond the scope of the review. Herein, we highlight some findings from representative studies that used clinical cohort data and machine learning-based approaches to classify depression subtypes.

## 3. Machine Learning-Based Symptom Clustering of Depression

Machine learning algorithms based on clinical data suggest clinical patterns for three to four subtypes of depression according to different studies. For instance, Kautzky et al. [34] examined 1079 patients with acute depression from a longitudinal multicenter study, conducted by the GRND. The researchers attempted to demonstrate the interplay between clinical and sociodemographic variables and their predictive influence on therapeutic outcomes. Hierarchical symptom clustering resulted in three subtypes of depression: emotional, anxious, and sleep- and appetite-related symptoms. Similarly, Chekroud et al. [35] used patient-reported data from the STAR*D trial (n = 4039) and applied hierarchical clustering to the Hamilton Depression Rating Scale (HAM-D) [36] and Quick Inventory of Depression Symptomatology Self-Report (QIDS-SR) items [37], thereby suggesting three clusters including “core emotional,” “atypical,” and “sleep” subtypes. Interestingly, the “emotional type” resembles the traditional melancholic subtype of depression, supporting the converging of data-driven classification of subtypes with theory-driven symptom grouping [38]. Furthermore, the symptom clustering findings from these studies are consistent with symptom-specific therapeutic responses. Whereas response rates for the clusters, including the “emotional” and “vegetative function” types, were comparable to the total HAM-D response rate, the rate for the anxiety cluster was significantly lower at 48.5% [34], indicating poor therapeutic response for anxious depression [39]. The poor response might be associated with the treatment side effects that manifest within the anxious cluster and can negatively impact any therapeutic effects on other symptoms. This mechanism was is in line with recent analyses of SSRI response [40]. Taken together, the recent clinical cohort studies resulted in similar symptom clusters that were driven by machine learning algorithms and agreed with clinical experience. Machine learning-based symptom clustering is presented in Table 1.

Symptom classification using machine learning algorithms should be conducted from the perspective of clinical experience and conventional theories [38]. Although symptom clustering is performed using learning algorithms without researcher intervention, the ratification of the resulting items is entirely based on researcher experience and knowledge. For instance, Chekroud et al. and Kautzky et al. similarly applied hierarchical clustering methods and named several resulting items “emotional clusters.” Both “core emotional clusters” include symptoms related to mood, energy, concentration, interest, and self-worth [41,42,43]. However, the emotional cluster by Kautzky et al. also included suicidality, whereas that by Chekroud et al. did not. In clinical practice, suicidality can occasionally be interpreted as an atypical feature of depression, rather than as a core emotional component, and may differ between atypical and other types of depression, according to previous analyses [44]. Other studies, however, have suggested that suicidality risk can be enhanced by the presence of anhedonia, which is regarded as a core emotional component [45,46]. The characteristics of suicidality congruent with emotional components are also supported by factorial analysis using HAM-D [47]. As the conventional concept of atypical depression involves hyperphagia and hypersomnia, further ratification of atypical clusters is required to adopt clinical scales such as QIDS-SR, rather than HAM-D, which does not assess the excess of vegetative symptoms. Meanwhile, contrary to the analysis results of Chekroud et al., anxiety symptoms were not included in the core emotional cluster but rather suggested to produce a separate cluster, along with psychomotor agitation, loss of sleep, and appetite. Independent appetite loss was also suggested in a previous factorial analysis [47]. The anxious cluster may be corroborated by studies that used GRND and GSRD subjects that recently established an anxious subtype of depression, in which anxiety symptoms were associated with somatic symptoms and generally poor therapeutic outcomes [39]. Taken together, data-driven symptom clustering can be further interpreted based on clinical experiences and ratified using the clinical consensus.

## 4. Exploring Prediction Model for Treatment Outcomes

In pharmacotherapy, personalized medicine holds promise in the treatment of depression, a heterogeneous disorder [48] for which no single antidepressant is identically effective, and for which numerous patients are given several treatments before a correct regimen is identified. At a population level, large-scale clinical trials, such as STAR*D and CO-MED, have shown that approximately 30% of patients achieve symptomatic remission for a given treatment [2,3]. However, personalized medicine shifts emphasis away from remission rates and treatment efficacy at the population level and attempts to identify the specific drug that is the best candidate for 30% of patients. For example, although general remission rates for all drugs were similar (48–52%), the analysis by Chekroud et al., which aimed to prospectively predict therapeutic outcomes, was more diverse (51–65%) [35]. Evidently, the development of generally effective antidepressants is crucial for public health. Until then, the application of pioneering statistical methods to choose the best drug candidate for each patient may offer an interim solution [49,50,51]. These findings are the first step in the era of personalized medicine for psychiatry. However, the performances are apparently inadequate in comparison with other areas of medicine.

Some representative studies have investigated methods to build prediction models for estimating antidepressant responses. Kautzky et al. [52] demonstrated that a random forest model for therapeutic response accurately identified 25% of melancholia responders by using three SNPs and clinical variables. Patel et al. showed that an alternating decision tree model estimated therapeutic response with an accuracy of 89% using age, structural imaging, and mini-mental status examination scores [53]. Chekroud et al. [35] demonstrated that a machine learning model forecasted remission with 59% accuracy using 25 clinical variables. Iniesta et al. [54] also showed that regularized regression models using demographic variables predict therapeutic response with clinically meaningful accuracy. Maciukiewicz et al. [55] recently reported that a support vector machine model predicted therapeutic response with 52% accuracy using SNPs. More precise prediction models, with accuracies of approximately 70%, have also been suggested for antidepressant therapeutic outcomes from large-scale clinical databases, including the GSRD and GRND in Europe, and the STAR*D cohorts in the US [30,31,32,56,57]. Table 2 describes some representative studies that have addressed the prediction of treatment response in depression.

The success of these models depends on their ability to be generalized. In general, there are two important precautions. First, all variable selection and model building should occur in the direction of ensuring validity. A vital challenge for prediction is determining the variables to be used. For example, Kautzky et al. [52] used a random forest procedure to identify 88 predictors that distinguish patients with unfavorable therapeutic outcomes from those with favorable outcomes, considering the number of interaction effects between predictive variables. Chekroud et al. adopted a penalized logistic regression with advantages: coefficients of correlated predictors are minimized toward each other, and uninformative features are uninvolved from the model [58,59]. They used the method to identify 25 predictive variables. Second, the way the model is trained on a large antidepressant cohort would be performed in other clinical trial cohorts with other treatment protocols, with different recruitment criteria and distributions of symptoms. An external validation analysis, for instance, showed that a citalopram model, trained in the STAR*D cohort, accurately predicted outcomes for the escitalopram treatment group of CO-MED. The model also showed significant prediction accuracy in the escitalopram-bupropion group, but not in the combination group of venlafaxine-mirtazapine [35]. This result shows that the model may generalize to an independent cohort sample and represent some caveats of therapeutic specificity. The finding that the model poorly predicts response to the venlafaxine-mirtazapine group indicates that the model does not predict a broad therapeutic response, nor does it predict equivalently for all treatment subgroups of CO-MED. The use of wholly independent validation cohorts also showed that although some predictors could still yield comparable model performance in the STAR*D escitalopram cohort, the model does not generalize to the escitalopram group of an independent clinical cohort, emphasizing the importance of external validation.

Further, statistical (and biomarker) approaches showed a performance above chance to be clinically usable. In STAR*D analyses, an accuracy of 53.13% would have surpassed the chance accuracy of 51.3% regarding the sample size, by the conventional statistics of *p* value < 0.05. The model by Chekroud et al. achieved an accuracy of 64.6%, outperforming this benchmark significantly. However, clinical experience-based predictions of who would respond to which therapeutics are generally poor [60]. Similarly, in a pilot sample of psychiatrists, the average accuracy of the 23 clinicians in predicting therapeutic outcomes for 26 STAR*D patients was 49.4%, given that the chance prediction was 53.9% [57]. These findings suggest that machine learning approaches can be a useful tool for predicting therapeutic outcomes with clinically meaningful accuracy.

## 5. Antidepressant Selection Specific to Symptom Clusters

Predicting who will respond to which treatment is a major challenge in personalized medicine. Machine learning algorithms are available tools for predicting differential responses to various antidepressants. The current research attempts to measure the extent of the differential predictive power of the algorithm approach for distinct medications. These efforts reflect the distinctive neural mechanisms probed in large-scale clinical cohort studies and heterogeneity in the underlying neurobiology of depression among the patients who entered the studies. Here, we address a recent representative study that explored antidepressant treatment outcomes according to each symptom cluster.

Chekroud et al. [35] used a citalopram-treated STAR*D cohort (n = 1962) with the aim of identifying differential response trajectories from three symptom clusters based on QIDS-SR scores. The unsupervised clustering approach yielded core emotional, atypical, and sleep/insomnia clusters among the citalopram STAR*D cohort. The CO-MED (n = 640) cohort was independently adopted as a test sample for the mixed-effects regression analysis. The CO-MED cohort included three subgroups of escitalopram (n = 151), escitalopram-bupropion (n = 134), and venlafaxine-mirtazapine (n = 140). Core emotional symptoms were predicted with significantly above-chance performance in the escitalopram and venlafaxine-mirtazapine groups. Sleep symptoms were predicted above chance for the escitalopram-bupropion group. These findings may translate into clinical applications. Escitalopram alone or a venlafaxine-mirtazapine combination may be beneficial for the core emotional symptoms related to energy/fatigability, concentration/decision-making, interest loss, sad mood, and feelings of worthlessness. Escitalopram-bupropion combination may have a beneficial impact on symptoms related to sleep onset insomnia, midnocturnal insomnia, and termination insomnia. In other words, no antidepressant was equally effective for all three symptom clusters, and for each symptom cluster, there were significant differences in treatment efficacy between drugs. In general, antidepressants worked best in treating the core emotional and sleep symptoms and showed less effectiveness in treating atypical symptoms. The extent of these differences suggests that selection of the best antidepressant for a given symptom allows for more benefit than that gained by the application of an active compound versus a placebo. Taken together, therapeutic outcomes at the symptom cluster level remain predictable using the machine learning algorithm of self-reported patient data.

These findings may help guide future research on personalized medicines for antidepressant selection. The finding by Chekroud et al.—better trajectories for core emotional symptoms with citalopram—is consistent with the findings of the genome-based therapeutic drugs for depression (GENDEP) study where symptom dimensions of cognition and mood were significantly better with escitalopram than with nortriptyline [23]. Whereas large-scale comparative studies of combined severity demonstrated modest differences between antidepressant classes [61,62], the results by Chekroud et al., at the symptom cluster level, indicate considerable differences between medications both within and across antidepressant classes.

Future clinical research should determine whether these clusters can be generalized to other clinical cohorts and reflect good candidates for a true depressive symptom structure [63,64]. The cluster structures by Chekroud et al. resemble those of other scales in other large samples of patients [61,65,66,67], although a recent review insisted that the argument is still ongoing [63]. These findings are largely consistent with those of Chekroud et al. in terms of independent sleep symptoms and core emotional symptoms, including sad mood, anhedonia, and self-worthlessness. However, the use of many different rating scales for depression impedes direct comparisons between cohorts [64].

## 6. Brain Imaging Techniques and Machine Learning in Depression

Machine learning algorithms are wildly applicable to diverse data of patients for elucidation of the complex nature of depression. In particular, in addition to the aforementioned clinical cohort data, there has been growing attention toward using brain imaging methods to detect endophenotypes of depression that can be clinically significant and feasible for translation to diagnosis [53,68]. Brain MRIs are one of the most widely used techniques that help identify the potential biological markers of depression. Importantly, MRI techniques with machine learning algorithms can produce a classification with brain networks and prediction of treatment response in depression. First, some representative studies adopted graph theory approaches [69,70,71,72] to detect defective functional and structural brain networks of depressed patients. Gong et al. [73] enumerated diverse brain network features, such as alterations in regional and connectivity patterns of different MRI modalities for depression, which include the regional betweenness and degree centrality in structural MRI, region-of-interest-based analysis in functional MRI, and white matter structural connectivity in the diffusion tensor image. Zeng et al. [74] examined the whole-brain functional connectivity at resting state to distinguish the depressed patients from the controls, which yielded 100% sensitivity. The most discriminant functional connectivity was found within or across the affective network, default mode network, and visual cortical areas, which seem to play a critical role in the neuromechanism of depression. Second, other representative studies sought to find alterations in brain network activity at resting state as potential endophenotypes for prediction of therapeutic outcomes. Drysdale et al. [75] suggested that four different patterns of fronto-striatal and limbic functional connectivity be defined as depression biomarkers from functional MRI analyses. The biomarkers were also related to distinctive profiles of clinical symptoms. For instance, biomarker 1 was associated with severe fatigue and anhedonia, and showed best response to repetitive transcranial magnetic stimulation. Redlich et al. [76] examined whether changes in gray matter volume predict response to electroconvulsive therapy. Support vector regression was accompanied with univariate analysis of the Hamilton Depression Rating Scale score, which successfully predicted the response to electroconvulsive therapy and reduction in HDRS. Jiang et al. [77] predicted remission after electroconvulsive therapy using the gray matter of depressed patients, in which six different gray matter networks were suggested as predictors of response to electroconvulsive therapy. Thus, the connectome-based endophenotypes may yield novel opportunities to define the diagnosis of depression and improve therapeutic response.

## 7. Future Research into Treatment Selection Models

We address the major steps involved in building antidepressant selection models from a clinical database that involves values, for each patient, on variables that represent clinical and demographic characteristics, therapeutics applied to the patient, and observed outcomes from the therapeutics. Understanding the sequential steps is crucial for interpreting and evaluating the utility of findings from the antidepressant selection studies.

The first step was to establish candidate predictor variables. Appropriate candidate predictor variables are those that are acquired prior to the treatment assignment and that credibility could be related to outcome, either generally or differentially between treatments. If a previous study has suggested that a variable can predict an outcome, then it should be involved as a potential predictor variable. However, as the literature on predictors of psychiatric disorders is still relatively scarce, considering other putative variables is recommended.

Variables should be free of significant missingness, and systematic missingness should be examined to ensure the appropriateness of imputation [78]. Variables should also show considerable variability. For instance, it does not make sense to involve sex if 90% of the sample is male. Selecting variables used for prediction is reliant on situations in which predictors exhibit high collinearity. Therefore, it is plausible to test the covariance structure of the putative predictors and take measures to reduce the high collinearity [78]. Other suggestions for identifying putative predictors include addressing outliers, making categorical variables binary, and converting variables for hypothetical reasons or handling highly skewed distributions.

Once putative predictor variables were selected, the next step was to construct the prediction model. This is typically a two-step procedure that includes variable selection and model specifications. Many different variable selection approaches have been suggested for treatment selection, all of which attempt to identify which variables, among the putative predictors, contribute significantly to the prediction outcome. Gillan and Whelan [79] presented an outstanding discussion of data-driven versus theory-driven approaches to model specifications. Typical approaches depend on parametric regression models [80] that select only variables with statistically significant contributions to the outcome. Another approach includes penalties with the goal of limiting the number of selected variables [81]. Others utilize bootstrapping processes that help maximize the generalizability of the models [58,82,83]. Progress in statistical modeling has led to feature selection methods, which are largely based on machine learning algorithms that can compliantly model and identify predictors, even with higher-order interactions [84]. Gillan and Whelan [79] provided an in-depth discussion of the merits of machine learning in the field of psychiatric disorders.

## 8. Conclusions

Research into depression is complicated by patient heterogeneity, which has motivated the search for homogeneous symptom clusters via data-driven computational approaches to identify predictor signatures in a cohort database. Along with the availability of clinical cohort data that are cost-effective and generalizable, machine learning algorithms enable symptom grouping and antidepressant selection. In particular, antidepressant selection in depression aims to help each patient receive the treatment, among the available options, which are most likely to yield a beneficial outcome for them. Symptom clusters putatively include core emotional, anxious, sleep/appetite, and atypical types, which seem congruent with the theory-driven grouping of depression, underlining that data-driven classification can agree with theory-driven approaches. Treatment outcomes differ in distinct antidepressant classes according to symptom clusters, although this pioneering finding should be replicated in other cohorts in the future. In addition, machine learning is a flexible platform to which brain imaging data can also be subjected to yield favorable performance in classification and therapeutics prediction in depression. Machine learning-based symptom clustering and antidepressant selection may be the first step in facilitating personalized medicine for better diagnostics and therapeutics for depression.

## Figures and Tables

**Table 1 diagnostics-11-01631-t001:** Data-driven symptom clustering of depression from clinical cohorts with over 500 depressed patients.

Cluster	Subtyping Based on	Relevant Phenotype	Statistical Method	Cohort	Study
Core emotional	QIDS-SR	Energy/fatigability		STAR*D (n = 4017)	Chekroud et al. [35]
	HAM-D	Concentration/decision making	Hierarchical clustering	CO-MED (n = 640)	Kautzky et al. [34]
		Loss of interest		GRND (n = 1079)	
		Mood (sadness)		GSRD (n = 1568)	
		Worthlessness			
Sleep	QIDS-SR	Sleep-onset insomnia	Hierarchical clustering	STARD*D (n = 4017)	Chekroud et al. [35]
	HAM-D	Mid-nocturnal insomnia		CO-MED (n = 640)	Kautzky et al. [34]
		Early-morning insomnia		GRND (n = 1079)	-
				GSRD (n = 1568)	
Atypical	QIDS-SR	Psychomotor agitation	Hierarchical clustering	STARD*D (n = 4017)	Chekroud et al. [35]
	HAM-D	Psychomotor retardation		CO-MED (n = 640)	
		Suicidal ideation			
		Hypersomnia			
		Hypochondriasis			
Somatic/anxious	HAM-D	Psychomotor agitation	Hierarchical clustering	GRND (n = 1079)	Kautzky et al. [34]
		Anxiety somatic		GSRD (n = 1568)	
		Anxiety psychic			
		Somatic symptoms			
		Genital symptoms			
		Hypochondriasis			
Appetite/weight	HAM-D	Appetite	Hierarchical clustering	GRND (n = 1079)	Kautzky et al. [34]
		Weight changes		GSRD (n = 1568)	

**Table 2 diagnostics-11-01631-t002:** Prediction models for treatment response in depression.

Study	Feature	Sample Size	Statistical Method	Prediction Accuracy on Treatment Response
Kautzky et al. [52]	three SNPs and clinical variables	225 depressed patients from GSRD	Random forest model	25% of melancholia
Patel et al. [53]	age, structual imaging, and mini-mental status examination scores	33 late-life depression patients and 35 non-depressed individuals	Linear and non-linear decision tree model	89%
Iniesta et al. [54]	clinical and demographic variables	793 depressed patients	Regularized regression model	72%
Chekroud et al. [35]	25 clinical variables	4039 depressed patients from STAR*D	Mixed-effects regression analysis	59%
Maciukiewicz et al. [55]	SNPs	186 MDD patients	Supporter vector machine model	52%
Kautzky et al. [30]	48 clinical, sociodemographic, and psychosocial predictors	400 training samples and 80 test samples	Random forest model	85%
Michael et al. [31]	HAMD-21 baseline score, episode length < 24 months and fewer previous hospitalizations	1014 naturalistically treated inpatients	Logistic regression and third-CART analyses	72%
Perlis et al. [32]	Clinical or sociodemographic variables	4039 depressed patients from STAR*D	Logistic regression model	71%
Kautzky et al. [34]	88 clinical predictors	1079 GRND samples	Random forest model	85%

## Data Availability

No further or supplementary data is stated in the study.

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
