# Peer review of "Machine Learning-Based Definition of Symptom Clusters and Selection of Antidepressants for Depressive Syndrome"

_diagnostics, 2021, doi:10.3390/diagnostics11091631_

Round 1

Reviewer 1 Report

The  topic of this paper is timely and of great importance.

The paper is well written.

It is a kind of review, but it is apparently not a systematic review in the modern sense following precise rules , but a  kind of narrative review focussing predominantly on papers which seem of predominant interest for the authors. This should be described and explained.

Although it is not a systematic review, it would be welcomed by the readers to know how many papers on this specific topic  are published und how the search was made, e.g. a kind of PRISMA diagram (which is a typical essence of a systematic review).

At some place the advantage of the machine learning approach compared to traditional methods like multivariate analysis should be explained. If available informative empirical examples comparing both methods would  be helpful for the understanding.

The description of the selected papers are well-done.

Author Response

It is a kind of review, but it is apparently not a systematic review in the modern sense following precise rules, but a  kind of narrative review focussing predominantly on papers which seem of predominant interest for the authors. This should be described and explained.

  • According to the kind advice, we inserted the description that the review is not a systematic or comprehensive, but rather narrative. (page 6)

At some place the advantage of the machine learning approach compared to traditional methods like multivariate analysis should be explained. If available informative empirical examples comparing both methods would be helpful for the understanding.

  • According to the kind advice, we inserted the explanation paragraph for the comparison between the machine learning and traditional methods. (page 5)

Reviewer 2 Report

This is an interesting review that highlights an important issue in psychiatry. The manuscript is well written, and the results are clearly presented. The Authors should better clarify the methods of the study selection. There are important studies on the argument that are missing such as "Pharmacogenomics-Driven Prediction of Antidepressant Treatment Outcomes: A Machine-Learning Approach With Multi-trial Replication; Arjun P Athreya et al 2019”

Author Response

This is an interesting review that highlights an important issue in psychiatry. The manuscript is well written, and the results are clearly presented. The Authors should better clarify the methods of the study selection. There are important studies on the argument that are missing such as "Pharmacogenomics-Driven Prediction of Antidepressant Treatment Outcomes: A Machine-Learning Approach With Multi-trial Replication; Arjun P Athreya et al 2019”

  • We appreciate the advice for the article inclusion in the revised manuscript. According to the advice, we included the article in the reference lists at number 8.